# Is (Disordered) Social Networking Sites Usage a Risk Factor for Dysfunctional Eating and Exercise Behavior?

**DOI:** 10.3390/ijerph20043484

**Published:** 2023-02-16

**Authors:** Lisa Mader, Kai W. Müller, Klaus Wölfling, Manfred E. Beutel, Lara Scherer

**Affiliations:** Outpatient Clinic for Behavioural Addictions, Department of Psychosomatic Medicine and Psychotherapy, The University Medical Centre of the Johannes Gutenberg University Mainz, Untere Zahlbacher Straße 8, 55131 Mainz, Germany

**Keywords:** social networking sites, social networks use disorder, active and passive usage, eating disorder, exercise dependence

## Abstract

Background: Research over the past years has shown that exposure to thin and beauty ideals in the media can be associated with disordered eating and related variables. Nowadays, interactive media, such as social networking sites, have gained growing popularity and represent a major part of people’s lives. It is therefore crucial to investigate how far users might be negatively influenced by social networking sites regarding eating pathology or excessive exercise behavior and if there are particular links to social media use disorder. Methods: Data were collected by an online-survey encompassing questions on regular social networking site use, eating disorders, and excessive exercise behavior. Results: Analyses showed that disordered social networking sites use was significantly related to eating pathology and a poorer body image in men and women. The frequency of active or passive social networking sites usage however was not associated with exercise behavior. Conclusions: Our results confirm that disordered social networking sites use represents a risk factor for body image dissatisfaction and associated eating disorders.

## 1. Introduction

Research of the past years has demonstrated that different sociocultural risk factors are associated with eating disorders. In particular, the exposure to thin and beauty ideals in mass media can be associated with negative body image, body dissatisfaction, as well as disordered eating, and that pertains especially for women [1,2,3,4,5,6,7]. Images of ideal thin-bodies have become highly existent in conventional media, such as fashion magazines and television. These images often convey an unrealistic beauty ideal of being young, tall, and extremely slim, so that a number of studies have been undertaken and many theories have attempted to explain the effects of such media content on eating pathology and related variables [2,8]. One of these theories is called sociocultural theory. According to this, women aspire to the above-mentioned beauty ideals, but cannot achieve them, resulting in body dissatisfaction. Further, idealistic images can cause women to internalize them and to compare themselves with them, again resulting in body dissatisfaction [9,10,11,12]. Another theory, called objectification theory, implies that consistent occurrence of beauty ideals in media lead women to self-objectification and body surveillance, what can cause increased body dissatisfaction and disordered eating behavior [13].

In the past years, social media and social networking sites (SNS), such as Facebook, Instagram, or Twitter, have become very popular. These sites occupy an essential part of young people’s lives [14,15]. Especially among young women, SNS are even more popular than conventional media formats [1,5]. Referring to this, it has been assumed that excessive use of SNS can result in problematic or even addictive behavior. Although not a category of its own, disordered SNS-usage (also called social-networks-use disorder; SNUD) can be subsumed under the category “other specified disorders due to addictive behaviours” in the ICD-11 [16]. Besides undoubted advantages, there are many problematic consequences related to SNS-usage, especially if it is related to addictive symptoms like tolerance, preoccupation, or loss of control. Further problems encompass decreases in sleep-quality, self-esteem, or well-being and an enhanced risk of further psychopathological symptoms [14,15,17,18]. Another scope that seems to be influenced adversely by intense to excessive SNS-usage is eating pathology and body image [8,19].

Associations were found between internet exposure and body image concerns in pre-teenage girls and female high school students. In detail, there are significant associations between internet exposure and internalization of the thin ideal, increased dieting, body surveillance, and reduced body esteem. These correlations were stronger for the time spent on SNS than for the time spent on the internet in general. Among a subsample of SNS-users, the number of online friends was correlated with drive for thinness, internalization of the thin ideal, and also body surveillance [20,21]. Likewise, Mabe and colleagues [22] found a correlation between more frequent SNS-usage and greater disordered eating in women. Other studies showed that increased photo activity on Facebook (e.g., posting, viewing, or commenting images) was significantly correlated with weight dissatisfaction, drive for thinness, thin internalization, and self-objectification in girls [23]. Moreover, McLean and colleagues [24] depicted that girls who regularly shared selfies on SNS, compared to those who did not, reported greater body dissatisfaction, thin-ideal internalization as well as over-evaluation of weight and shape. Additionally, higher investment in posted photos was correlated with higher body-related and eating concerns in girls who often posted selfies. Correspondingly, Cohen, Newton-John and Slater [25] found that in comparison to overall SNS-usage, the appearance-focused SNS-usage was related to body image concerns and thin-ideal internalization in young women. These findings imply that not all SNS-activities are linked to body image concerns in the same way. As SNS are apparently not homogenous and allow individuals various types of activities (e.g., viewing and commenting photos or videos, uploading photos or stories, chatting etc.) it is necessary to examine specific patterns of internet use rather than simply address the total time spent online, to gain better understanding of related outcomes. The relevance of considering specific usage types can also be accentuated by the fact that some types of internet usage are associated with fewer symptoms of depression, while other types are linked to increased depressive symptoms. Therefore, it is also required to examine if relationships between SNS-usage and body image variables depend on specific usage types [14,25,26,27].

With regard to content posted on SNS, it was shown that in comparison to women who post travel pictures on Instagram, women posting so-called “fitspiration” images scored significantly higher on maladaptive eating and exercise behavior variables like drive for muscularity and compulsive exercise [28]. This meets findings on the coexistence of compulsive exercise behavior and the emergence or maintenance of eating disorders [28,29]. It also corresponds to previous findings from [30,31] showing that exposure to thin and athletic ideal images influence body dissatisfaction. Indeed, it has even been illustrated that exposure to athletic images leads to greater body dissatisfaction compared to exposure to muscular ideal images or traditional thin ideal images. However, exposure to fitness images led to increased inspiration to exercise but did not affect actual exercise engagement [31].

Another point is that further experimental research demonstrated that short confrontation with Facebook can induce negative mood states and weight/shape concerns [19,22]. Similarly, exposure to fitspiration images caused increases in negative mood and body dissatisfaction as well as reduced self-esteem in women [32]. Further, Brown and Tiggemann [33] showed that exposure to celebrity and unknown attractive peer images leads to increased body dissatisfaction and negative mood whereas travel images did not have this effect. Likewise, facial dissatisfaction is affected positively too by thin-ideal images on Instagram [34]. Casale and colleagues [35] demonstrated that exposure to appearance-focused and attractive same sex, real Instagram profiles for a period of one week, led to increased body dissatisfaction in female participants. The exposure caused greater definition of themselves and their self-worth through physical appearance in female, but not in male participants.

### Aim of the Current Research

The aim of the present pilot study was to analyze the relationship between SNS-usage, eating, and exercise behavior in order to confirm and extend existent findings [20,21,22,28]. We predicted a positive correlation of addictive SNS-usage with disordered eating and excessive exercise behavior for men and women. Moreover, we aimed at extending previous findings [23,24,25] by exploring, if there are relationships between the type of SNS-usage and disordered eating, related body image concerns, and exercise behavior. As SNS allows various types of active and passive usage, we explored if the frequency of these usage-types is associated with excessive exercise and disordered eating as well as related body image variables. This approach seems also relevant following findings of Holland and Tiggemann [28] who demonstrated that women who posted fitspiration images on Instagram—to be considered as active SNS-usage—scored significantly higher on maladaptive eating and exercise behavior. However, a passive consumption of fitspiration images may not fulfil the intended purpose of higher engagement in exercise, as Robinson and colleagues [31] showed in their laboratory setting. 

Further we aimed at exploratively analyzing further predictors of eating and exercise behavior such as age and sex, important variables in the context of SNS-usage [1,5] as well as in eating and exercise behavior [5,25,28]. Lastly, we put emphasis on gender-specific analyses for all our research questions in order to determine potential and currently under-researched differences between women and men.

## 2. Materials and Methods

### 2.1. Participants and Procedure

After obtaining approval from the local Ethics Committee, we started recruitment of participants. The sample was recruited from the general population through different SNS posts, printed ads in the region, as well as within the patient care in an outpatient clinic of the University Medical Center. A self-rated regular SNS-use was defined as the only inclusion criterion. All participants completed an online-survey and were assessed for study relevant constructs (SNS-usage, eating and exercise behavior). Participants were n = 71 women and n = 51 men aged between 12 and 61 years (M = 25.85; SD = 6.86). Regarding family status, 48.4% were in a relationship, 43.4% were single, and 6.6% married. Most of them were students (63.1%), 29.5% were employed and the rest job-seeking or homemaker. Regarding educational level, 54.1% reported to have “Abitur/Fachabitur” (A-level/high educational level), 32.8% a university degree, 5.7% “Realschulabschluss” (medium educational level), 4.1% “Hauptschul-/Volksschulabschluss”(low educational level) and 3.2% with no graduation, other graduation, or still at school.

### 2.2. Measures

The online-survey questionnaire contained questions about general socio-demographic data and specific study relevant content (see below).

### 2.3. Social Media Disorder Scale (SMDS)

We used the short version of the social media disorder scale [36] to assess symptoms of social-networks-use-disorder. It classifies social media users in disordered (i.e., addicted) and non-disordered users (with a cut-off of 5). It consists of 9 dichotomous items (0 = “no”, 1 = “yes”) covering crucial diagnostic criteria that have been defined for the overarching construct internet addiction. So, a higher sum value indicates more pronounced SNUD-symptoms. Psychometric properties have been successfully evaluated and, in our survey, the internal consistency (Cronbachs α) was α = 0.71.

### 2.4. Type of Social Media Usage

To assess the frequency of the social media usage-types, participants were asked how often they use SNS in an active way, by creating own content (posting etc.) and how often they use SNS in a passive way, by consuming content of others (browsing through profiles of others, watching postings of others etc.). Responses were assessed on an 8-point-frequency-scale (never, less than once a month, once a month, twice to thrice a month, once a week, six times a week, daily, several times a day).

### 2.5. Eating Disorder Examination Questionnaire (EDE-Q)

The EDE-Q [37] operationalizes key attitudes and behavioral features of disordered eating over the past month. It comprises 28 items, whereof 22 items acquire specific psychopathology of disordered eating through the 4 subscales “Restraint”, “Eating Concern”, “Weight Concern” and “Shape Concern”. A further 6 items acquire diagnostic relevant key behavior. It is a self-report questionnaire, and each item has to be rated on a 7-point forced-choice scheme. The questionnaire can differentiate between people with and without eating disorder. We used the German version of the EDE-Q [38], which has been successfully evaluated and showed good psychometric properties [39]. In our sample the total consistency was α = 0.94 (for subscales: Restraint: α = 0.83, Eating Concern: α = 0.79, Weight Concern: α = 0.85; Shape Concern: α = 0.91).

### 2.6. Exercise Dependence Scale-21 (EDS-21)

The Exercise Dependence Scale-21 (EDS-21; [40]) is a self-reporting questionnaire to assess exercise dependence based on the diagnostic criteria for substance dependence defined in the Diagnostic and Statistical Manual of Mental Disorder–IV (DSM-IV). It consists of 21 items, whereof three items represent one of seven subscales/factors based on the diagnostic criteria. Each item has to be rated on a 6-point Likert scale. The questionnaire distinguishes between individuals who are at-risk for exercise dependence, nondependent-symptomatic, and nondependent-asymptomatic. We used the German version of the EDS, which has been successfully validated and has shown sound psychometric properties [41]. In our sample the total internal consistency was α = 0.96 (for subscales: Tolerance α = 0.9, Withdrawal α = 0.75, Intention Effects α = 0.90, Lack of control α = 0.92, Reduction in other activities α = 0.71, Time α = 0.93, Continuance α = 0.8).

### 2.7. Data Analyses

SPSS 23.0 was used for all statistical analyses. After testing and approving the relevant statistical requirements, correlational analyses (bivariate Pearson-correlations (r)) and multiple linear regression analyses were performed for the examination of relationships of the variables. To investigate group differences, non-parametric tests were applied.

## 3. Results

The vast majority of our sample (98.4%) indicated use of the internet in their leisure-time on a daily basis. The number of 38.5% reported use of the internet of less than one hour per day, 30.3% used it 1–2 h per day, 20.5% used it 2–4 h daily, and the rest spent more than 6 h. The majority (63.1%) reported use of SNS in a passive way (consuming content of others) daily, while only 7.5% reported use of SNS in an active way (posting own content) daily. 

In our sample, n = 8 participants exceeded the SMDS cut-off (≥5) which corresponds to a prevalence rate of 6.6% for SNUD (female: 8.5%; male 3.9%). The means for each item of the SMDS can be found in Table 1.

We found a prevalence of 2.5% (n = 3) for exercise dependence (female: 1.4%; male: 3.9%), 40.2% (n = 49) were classified in the symptomatic non-dependent group (female: 45.1%; male: 33.3%), and 57.4% (n = 70) were asymptomatic non-dependent (female: 53.5%; male: 62.7%). The number of 4.1% (n = 5) scored above the threshold for dysfunctional eating behavior (female: 4.2%; male: 4.0%).

While prevalence rates did not differ significantly between sexes (SMDS: χ^2^ = 2(1) = 0.994; *p* = 0.319; EDEQ: χ^2^ = 2(1) = 0.004; *p* = 0.951; EDS-21: χ^2^ = 2(2) = 2.22; *p* = 0.329), significant sex differences in all EDEQ-scores were found (see Table 2).

### 3.1. Relationship between Overall Social Networking Use Disorder-Symptoms, Eating Behavior, and Body Image

We found strong and positive correlations between SNUD-symptoms, eating behavior and body image variables. In detail, each subscale (Restraint, Eating Concern, Weight Concern, and Shape Concern) and the global score of EDE-Q correlated significantly with the SMDS-score (cf. Table 3). After separation for sex, there were no remarkable changes in these correlations. 

### 3.2. Relationship between Overall Social Networking Use Disorder-Symptoms and Exercise Behavior

No relationship between SNUD-symptoms and exercise behavior was found. Neither the total score of the exercise dependence scale nor one of its subscales correlated significantly with the social media disorder scale (for details see Table 3). After separation into sex, we found small but significant correlations in the subscales “Lack of Control” (r = 0.33, *p* = 0.02) and “Reduction in Other Activities” (r = 0.31, *p* = 0.03) for men only. For women, the correlations remained non-significant.

### 3.3. Relationship between Active and Passive SNS-Usage, Eating, and Exercise Behavior

No significant correlations between active (r = 0.10, *p* = 0.28) or passive SNS-usage (r = 0.13, *p* = 0.14) and eating behavior and related variables (EDE-Q) were found. Further, there were no significant correlations between active (r = −0.01, *p* = 0.96) or passive SNS-usage (r = 0.01, *p* = 0.9) and exercise behavior (EDS-21). After segregation for sex, the correlations remained non-significant.

### 3.4. Results of Multiple Linear Regression Analyses

Additionally, two multiple linear regression analyses were performed. Age, sex, and SMDS-score entered the model to predict EDE-Q-score and EDS-score respectively. For EDE-Q a significant model was identified (R^2^ = 0.307, *p* < 0.001) with age (β = 0.165, *p* = 0.035) and SMDS-score (β = 0.468, *p* < 0.001) as significant predictors (cf. Table 4). The second model on EDS, however, failed to show significant relationships (*p* = 0.795).

## 4. Discussion

In this pilot study, we investigated the relationships between SNS-usage, SNUD-symptoms, eating behavior, related body image variables, and exercise behavior. We were able partially to replicate previous findings from the literature in different respects, e.g., a higher prevalence rate of SNUD among women [18]. As predicted, SNS-usage turned out to be a significant predictor of eating pathology and SNUD-symptoms were strongly related to disordered eating and body image related variables, which is consistent with many results in the literature [20,21,22,23]. In detail, we found correlations between restraint eating behavior, greater eating-, weight- and shape-concerns, which indicates that a more intense SNS-usage also comes along with greater body and weight dissatisfaction and a higher dependence of the self-esteem from shape and weight. These results seem to fit the framework of the sociocultural theory, if we assume that participants were exposed to beauty ideal images on SNS, which actually is highly probable because of the wide distribution of these images on SNS. In that case, frequent confrontation with unrealistic beauty ideals in individuals suffering from SNUD, encourages internalization and comparison with them and to aspire to these ideals, but they are mostly not achievable. Both are possible mechanisms leading to body dissatisfaction, which in turn plays an important role in the context of disordered eating.

Not inferred from our correlational results but concluded from and connected to experimental studies [32,33,35] we propose that a more intense SNS-usage, especially when related to addictive use, leads to higher body image concerns and affects eating behavior adversely. In this case it means that SNUD-symptoms might represent a risk factor for more restraint eating behavior and greater eating-, weight-, and shape-concerns.

Based on previous results [23,25] showing that particularly specific SNS activities are linked to body image variables, we aimed at extending existing results with our study in which we investigated SNS-usage in a more nuanced way. We examined the frequency of active and passive SNS-usage in conjunction with eating and exercise behavior but were not able to find any associations. At first, this indicates that such ways of usage may not have an influence on eating and exercise behavior. However, maybe it would be worthwhile to have a closer look at this aspect while defining the active and passive usage more precisely, as for example questioning the exact type of activity that is most commonly shown. Further, as this was a pilot study with a small total sample and especially a small group of active users (7.5%), it is of interest to have a closer look at these usage-types in future research.

This study confirmed and partially extended existent results by examining the described associations in a sample of men and women with self-assessed intense to disordered social media-usage, while most of the literature in this research area is focused on the female population. Although we found higher scores of restraint eating, eating-, weight- and shape concerns for women, that fit existent literature, surprisingly we found almost equal prevalence rates for disordered eating for both sexes. This is not consistent with most epidemiological research, which has shown that eating disorder pathology is higher among women [38,42]. Consequently, we found all the same strong correlations between disordered SNS-usage, disordered eating, and related variables after separating for sex. This may be an indication that the mechanisms described within the sociocultural theory also apply for men. Therefore, the male population should not be ignored in this field of research.

Against our expectations we could neither identify SNUD-symptoms as a predictor of exercise dependence nor find any significant correlation between SNUD-symptoms and exercise behavior. These results did not fit our expectations derived from existent results that “fitspiration-Blogger” scored significantly higher on compulsive exercise behavior [8]. This may be due to the fact that we only found a few correlations between exercise and eating behavior scales in our sample, that do not fit existent literature as well [41]. After regarding these correlations segregated for sex, we found that SNUD-symptoms come along with “lack of control over exercise behavior” and a “reduction in other activities” only in men. This is an interesting finding because most previous studies have focused on the female population or found associations only for women. Likewise, Casale and colleagues [35], who investigated both sexes, did not find significant effects for men. Thus, it is essential to address this point in future research to gain better understanding and obtain clarity.

Our findings have to be interpreted in the light of several limitations. As this is a pilot study based on associations with SNS-usage types and SNUD-symptoms, we have to consider the small total sample size restricting the validity and possibilities on statistical analyses of our results. Even so the small sample size of the active users has to be seen as a limitation for the same reason. In future research we should make sure that the groups of active and passive users have equal group sizes to ensure sound group comparisons. Moreover, the measures of internet, SNS-usage, and SNUD-symptoms were based on self-reports and the recruitment of people with intensive SNS use was self-assessed, which in general can lead to distortions. As individuals define intense usage differently, this can be a possible explanation for the fact, that we have a few intensive SNS users with a usage time of less than one hour per day. This is also another point that can lead to distorted results. Furthermore, the present study followed a correlational design which means that causal conclusions cannot be drawn. So basically, it is possible that SNS exposure affects eating behavior and related body image variables, but the reverse is also plausible. Yet, in the light of previous experimental studies [32,33,35] it seems at least likely that SNUD-symptoms affect eating behavior and body image variables.

Eventually the present results give rise to further and more specific examinations in this field of research. Especially gender aspects as well as age should be taken into account, because we identified age as a significant predictor of eating pathology and to our knowledge existing studies focused only on younger women. Further experimental and longitudinal designs are needed to better understand the underlying processes and to acquire knowledge on the long-term effects.

Moreover, these findings provide an opportunity to derive important practical and clinical implications:(1)One of them refers to promoting a more conscious and critical dealing with SNS-usage and the content consumed, such as idealized images. These aspects can be implemented in the treatment and prevention of eating disorders primarily through informing patients about the fact that an addictive SNS-usage can be a risk factor for eating disorders and may worsen such symptoms.(2)In the light of the strong influence of SNS-usage, another important implication could be the consumption of SNS content that leads to better mood and body satisfaction as Slater and colleagues [5] demonstrated. Their experimental study demonstrated that women who viewed self-compassion images showed higher body satisfaction and appreciation as well as a reduction in negative mood in contrast to women exposed to neutral pictures. On the one hand this could provide the opportunity to reduce the negative impact of SNS on body image variables and on the other hand it could boost a positive effect of SNS on body image variables. These are important issues especially in the context of protection from eating pathology. So, it is important to take care about the content we consume in a passive way and the content we produce actively on social network sites.

## 5. Conclusions

In conclusion, the present study demonstrates the important role that the internet, especially social networking sites have in our lives nowadays and that SNUD-symptoms are associated with heightened body image concerns and disordered eating in men and women. Based on these results it can be assumed that disordered SNS-usage might represent a relevant risk factor in the context of developing or maintaining eating disorders. For the moment, it appears that there is no difference between active and passive usage of SNS relating to eating and exercise behavior. As SNS-usage has become such an essential part of our lives, future research is of particular relevance.

## Figures and Tables

**Table 1 ijerph-20-03484-t001:** Means of the social media disorder scale items for both sexes.

Criterion	SMDS-Item	Total (N = 122) M (SD)	Women (N = 71) M (SD)	Men (N = 51) M (SD)
	**During the past year, have you…**			
Preocccupation	…regularly found that you can’t think of anything else but the moment that you will be able to use social media again?	0.10 (0.30)	0.14 (0.35)	0.04 (0.19)
Tolerance	… regularly felt dissatisfied because you wanted to spend more time on social media?	0.11 (0.31)	0.13 (0.34)	0.08 (0.27)
Withdrawal	… often felt bad when you could not use social media?	0.11 (0.31)	0.11 (0.32)	0.10 (0.30)
Persistence	… tried to spend less time on social media, but failed?	0.26 (0.44)	0.35 (0.48)	0.14 (0.35)
Displacement	… regularly neglected other activities (e.g., hobbies, sport) because you wanted to use social media?	0.22 (0.42)	0.31 (0.47)	0.10 (0.30)
Problem	… regularly had arguments with others because of your social media use?	0.07 (0.26)	0.09 (0.29)	0.06 (0.24)
Deception	… regularly lied to your parents or friends about the amount of time you spend on social media?	0.07 (0.25)	0.07 (0.26)	0.06 (0.24)
Escape	… often used social media to escape from negative feelings?	0.35 (0.48)	0.52 (0.50)	0.12 (0.33)
Conflict	… had serious conflict with your parents, brother(s) or sister(s) because of your social media use?	0.05 (0.22)	0.07 (0.26)	0.02 (0.14)

Note. SMDS = social media disorder scale; M = mean; SD = standard deviation.

**Table 2 ijerph-20-03484-t002:** Gender-specific differences in body-image concern variables and the total score of the eating disorder examination questionnaire.

EDE-Q-Dimension	Women (N = 71) M (SD)	Men (N = 51) M (SD)	Singificance
Restraint	1.87 (1.56)	1.10 (1.27)	U = 1206.5, *p* = 0.003
Eating Concerns	0.85 (1.00)	0.54 (1.05)	U = 1139.5, *p* = 0.001
Weight Concerns	2.12 (1.60)	1.22 (1.32)	U = 1171.5, *p* = 0.001
Shape Concerns	2.44 (1.65)	1.64 (1.41)	U = 1246.0, *p =* 0.005
EDE-Q total score	1.82 (1.26)	1.12 (1.10)	U = 1161.5, *p* = 0.001

Note. EDE-Q = eating disorder examination questionnaire; M = mean; SD = standard deviation.

**Table 3 ijerph-20-03484-t003:** Prevalence of Social Media Addiction, Disordered Eating and Exercise Dependence, and Related Means (SD) and correlations between intensity of social networking sites usage (SMDS), Eating Behavior, Restraint Eating, Eating-, Weight- and Shape Concerns (EDE-Q), and Exercise Behavior (EDS-21).

	M	SD	α	Prevalence	1	2	3	4	5	6	7	8	9
(1) sex					1								
(2) age	25.85	6.86			0.01	1							
(3) SMDS	1.34	1.68	0.71	6.6%	0.32 **	0.05	1						
(4) EDE-Q global score	1.53	1.24	0.94	4.1%	0.28 **	0.19 *	0.52 ***	1					
(5) EDE-Q Restraint	1.55	1.49			0.26 **	0.14	0.32 ***	0.77 ***	1				
(6) EDE-Q Eating Concern	0.72	1.03			0.15	0.16	0.45 ***	0.83 ***	0.48 ***	1			
(7) EDE-Q Weight Concern	1.75	1.55			0.29 **	0.19 *	0.51 ***	0.93 ***	0.55 ***	0.76 ***	1		
(8) EDE-Q Shape Concern	2.11	1.60			0.25 **	0.17	0.52 ***	0.95 ***	0.61 ***	0.73 ***	0.92 ***	1	
(9) EDS-21	45.12	21.48	0.96	2.5%	0.02	0.06	0.05	0.1	0.25**	0.1	0.03	−0.01	1

Note: N = 122; *** *p* < 0.001; ** *p* > 0.05; * *p* < 0.01; α = Cronbach’s Alpha; SMDS = social media disorder scale; EDE-Q = eating disorder examination questionnaire; EDS-21 = exercise dependence scale-2.

**Table 4 ijerph-20-03484-t004:** Results of the linear regression analyses with age, gender, and symptoms of social networking use disorder on eating disorder symptoms.

	B	SE B	*ß*
constant	−0.201	0.481	--
age	0.030	0.014	0.165 *
Gender	0.318	0.127	0.127
SMDS-score	0.343	0.468	0.468 ***

Notes. B = regression coefficient; SE B = standard error of B; *ß* = standardized beta coefficients; * *p* < 0.05; *** *p* < 0.001.

## Data Availability

The data presented in this study are available on request from the corresponding author. The data are not publicly available due to reason of data protection.

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
