# Peer review of "Is (Disordered) Social Networking Sites Usage a Risk Factor for Dysfunctional Eating and Exercise Behavior?"

_ijerph, 2023, doi:10.3390/ijerph20043484_

Round 1

Reviewer 1 Report

1. I think this paper only confirms the relationship between Social Media Disorder and health, and the abstract states "Our results confirm that social networking sites overuse represent a risk factor for body 19 image dissatisfaction and associated eating disorders." I don't think overuse and Social Media Disorder are the same concept. Could the author please give me the appropriate explanation.
2. regression analysis need to add more detailed steps, mention the table that can contain detailed information.
3. I am confused because the author's data analysis only focuses on correlation analysis so far and the regression analysis is very brief. So I think it is not appropriate to express the causal relationship between variables more in the discussion. So I suggest that the authors revisit the correlation between the data analysis and the discussion and think about whether the results of the data analysis can be stated in terms of causality.
4. I would like to ask the authors to list the theoretical contributions, practical contributions, and shortcomings of this paper and future research directions separately, not together.

Author Response

(1) I think this paper only confirms the relationship between Social Media Disorder and health, and the abstract states "Our results confirm that social networking sites overuse represent a risk factor for body image dissatisfaction and associated eating disorders." I don't think overuse and Social Media Disorder are the same concept. Could the author please give me the appropriate explanation.

Thank you for this suggestion. Indeed, here is a need for precision. We have changed the sentence accordingly (“Our results confirm that disordered social networking sites use represent a risk factor for body image dissatisfaction and associated eating disorders.”). Please also note that – for more clarity – we changed the phrase “intense SNS-usage” into SNUD-symptoms throughout the manuscript.

(2) regression analysis need to add more detailed steps, mention the table that can contain detailed information.

We have now added a new table (table 4) to provide more detailed information on the regression analysis results.

(3) I am confused because the author's data analysis only focuses on correlation analysis so far and the regression analysis is very brief. So I think it is not appropriate to express the causal relationship between variables more in the discussion. So I suggest that the authors revisit the correlation between the data analysis and the discussion and think about whether the results of the data analysis can be stated in terms of causality.

We fully agree with your concern on potential causalities. Particularly since the data have been derived from a cross-sectional design, we have now rephrased essential parts of the discussion section and provided more cautious interpretations of the results.

(4) I would like to ask the authors to list the theoretical contributions, practical contributions, and shortcomings of this paper and future research directions separately, not together.

Thank you. Please note that we made several changes in the discussion section of the revised version. We hope that the structure of this part has gained in quality and comprehensibility.

Reviewer 2 Report

The topic of the paper is exciting.

I have few concerns:

1.       Authors should include a section after the introduction called “Literature Review.” To provide sufficient background for the studies already done and what the findings are.

2.       Authors should point out implications separately.

3.       Authors should indicate Limitations and Future Studies.

4.       Lines no. 198 to 206 looks very clumsy. The authors should rewrite this part.

5.       Authors may provide a Table to point out the findings more clearly.

6.       Lines no. 134 to 136, it should be (.), not (,). Authors must be more careful.

7.       Page No. 8 Line no. 77, Like all studies, our findings have…

Authors must provide references. 

Author Response

(1) Authors should include a section after the introduction called “Literature Review.” To provide sufficient background for the studies already done and what the findings are.

Thank you for that suggestion. In the original version, we closely related to the template provided by IJERPH. However, in order to improve the transparency for the readers, we have now added a separate paragraph “Aim of the current research” within the introduction section.

(2) Authors should point out implications separately.

Thank you. We have now outlined the two major implications separately.

(3) Authors should indicate Limitations and Future Studies.

We have now re-phrased this passage and provide further elaborations on the limitations and possible future directions of research on this topic.

(4) Lines no. 198 to 206 looks very clumsy. The authors should rewrite this part.

Thank you, we have now inserted a new table to make it more clear to the reader.

(5) Authors may provide a Table to point out the findings more clearly.

Thanks. A new table (table 2) has been inserted now.

(6) Lines no. 134 to 136, it should be (.), not (,). Authors must be more careful.

Thank you. We have now corrected this aspect throughout the manuscript.

(7) Page No. 8 Line no. 77, Like all studies, our findings have… Authors must provide references. 

Thanks again, you are absolutely right. Please note that we re-phrased the whole paragraph. In the original version, this paragraph led to a misunderstanding. We hope that we have now found a way to make it clearer.

Round 2

Reviewer 1 Report

Thank you for your response